# Climate Change, Pressures, and Adaptation Capacities of Farmers: Empirical Evidence from Hungary

**József Lennert** [1,*], **Katalin Kovács** [1], **Bálint Koós** [1], **Nigel Swain** [2], **Csaba Bálint** [3], **Eszter Hamza** [3], **Gábor Király** [3], **Katalin Rácz** [3], **Monika Mária Váradi** [1] and **András Donát Kovács** [1]

1   Institute for Regional Studies, HUN-REN Centre for Economic and Regional Studies, Tóth Kálmán u. 4, 1097 Budapest, Hungary; kovacs.katalin@krtk.hun-ren.hu (K.K.); koos.balint@krtk.hun-ren.hu (B.K.); varadi.monika@krtk.hun-ren.hu (M.M.V.); kovacs.andrasdonat@krtk.hun-ren.hu (A.D.K.)
2   Department of History, University of Liverpool, Liverpool L69 3BX, UK; swainnj@liverpool.ac.uk
3   Institute for Agricultural Economics, Zsil u. 3-5, 1093 Budapest, Hungary; balint.csaba@aki.gov.hu (C.B.); hamza.eszter@aki.gov.hu (E.H.); kiraly.gabor@aki.gov.hu (G.K.); racz.katalin@aki.gov.hu (K.R.)
*   Correspondence: lennert.jozsef@krtk.hun-ren.hu

**Abstract:** This paper aims to analyze comprehensively the climate exposure, sensitivity, perception, adaptive capacity, vulnerability, and resilience of the Hungarian agricultural sector, particularly focusing on fruit, vegetable, and grape producers. Four distinct Hungarian case studies were examined, representing different regions with diverse environmental and socioeconomic conditions. The research combined quantitative and qualitative methods, including statistical and GIS analysis of climate, agricultural, and socioeconomic data, as well as field research and semi-structured interviews. The study investigated exposure, sensitivity, perception, and adaptation, leading to the identification of key components and influencing factors. Qualitative research revealed that farms operating in geographically close proximity, in the same regulatory and support environment, can have different adaptive capacities. In the current state of the adaptation process, the extent to which farmers can rely on supportive professional networks and seek out and adopt new practices is crucial. Field experience suggests that without a strong and supportive producer organization (extension network), farmers may prefer to resort to extensification (afforestation) to mitigate production risks. From a development policy perspective, it is worthwhile to present good practices and provide information on possible adaptation techniques through existing local sectoral organizations.

**Keywords:** climate change; Hungary; horticulture; field research; exposure; sensitivity; perception; adaptive capacity; vulnerability; resilience



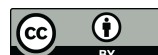

## 1. Introduction

It took 129 years for scientific consensus to develop on the impact of atmospheric carbon dioxide on the climate. The first scientific paper on the greenhouse effect of atmospheric carbon dioxide was published in 1856; it also suggested that changes in the amount of carbon dioxide could affect climatic conditions [1]. Although research continued, it was not until the mid-1980s that a scientific consensus on the issue emerged. An important milestone was the 1985 World Meteorological Organization (WMO) conference which confirmed a clear link between greenhouse gases and climate change, building on the results of global climate modeling [2]. Since then, there has been significant progress in the development of global climate models, with improved data collection and processing leading to more refined global and (later) regional climate models.

The first global climate models only predicted the Earth's average temperature, but building on these results, many research projects were launched to map the likely impacts of climate change [3]. The initial results were alarming, with food shortages, masses of farmers losing their livelihoods, cities flooded by seawater, and critical infrastructure becoming inoperable. A common feature of these studies was that they ignored adaptation by the

actors concerned, inevitably assuming that without substantive change, expected climate change would have such catastrophic consequences. These first alarming results triggered a great deal of scientific interest, resulting in a number of scientific disciplines becoming involved in mapping possible impacts, using their own terminology and methods. Five concepts—exposure, sensitivity, vulnerability, adaptability, and resilience—have gained significance in recent times. However, it is vital to acknowledge that they are not exhaustive, and their understanding is not necessarily uniform. As the issue of adaptation to climate change arises, we are faced with such a lack of knowledge about the typical climate adaptation intentions of actors, their direction, timing, perceived constraints (etc.), that we must now focus on the level of the decision-makers involved. It is particularly important to explore individual perceptions and adaptation intentions when there is a high degree of uncertainty about the likely local impacts of climate change.

Although climate change is a universal phenomenon, its effects are far from uniform, as topography, hydrology, etc., can have a significant impact on the process. This is particularly true in areas located far from the coast, such as Central Europe and Hungary, where the various effects are mixed, increasing the variability of the weather. A large number of climate models were developed to model the weather in Europe at the end of the 20th century, but their results are rather scattered: while there is little variation in temperature increases across models, there are marked differences in the expected precipitation and its temporal distribution (for more details see [4–6]).

Agriculture, one of the sectors most affected by climate change, has been at the center of climate change adaptation research from the very beginning, typically looking at impacts on arable crops (wheat, maize) and the potential and willingness to adapt. It is only in recent years that the impact of climate change on orchards and vineyards, as well as farmers' intentions and constraints to adapt, have come to the fore. On the one hand, the fruit and wine sectors are closely linked to several other sectors (food industry, tourism) and play a very important role in employment, especially in less developed rural areas. On the other, fruit and vine growers, who have built up and stabilized their businesses over the last three decades, are now faced with the dual and increasingly pressing challenges of climate change and labor shortage, which have changed farming conditions. In the case of vine and fruit farmers, it is therefore of particular importance to take a complex approach to examining their climate exposure, sensitivity, and adaptive capacity.

Keeping these considerations in mind, the objectives of our study are as follows:

- To analyze the climate exposure, sensitivity, perception, adaptive capacity, and adaptation practices of the Hungarian agricultural sector, with a specific emphasis on fruit, vegetable, and grape producers.
- To identify the relevant components and influencing factors associated with exposure, sensitivity, perception, and adaptive capacity.
- To investigate differences in the spatial manifestation of these concepts through the examination of four case studies.
- To assess the climate vulnerability and resilience of the agricultural systems in the four designated case districts.
- At the same time, the practical aim of the research is to provide Hungarian development policy with well-founded information on the most important labor-intensive agricultural sub-sectors, how they perceive climate change, how they assess the need for adaptation, and possible directions for it. An important aspect here is whether it is possible to rely on centralized information or whether it is more appropriate for public policy to support actors' information gathering and learning processes.
- After this introductory section, the remainder of this paper has the following structure. In Section 2, we provide a theoretical framework of the concepts relevant to the analysis, as well as a literature review. The applied methodology and data sources used are summarized in Section 3. In Section 4, we provide a description of the four case studies. Section 5 presents the results of the qualitative and quantitative analysis for climate change exposure, sensitivity, perception, adaptation, and adaptive

capacity. Finally, Section 6 summarizes the results at the case study level: it provides an assessment of climate vulnerability and resilience for the case studies, highlights the most important findings, discusses the results in light of the earlier findings, presents the limitations of the study, and suggests future research directions.

## 2. Theoretical Framework and Literature Review

### 2.1. Theoretical Framework

Understanding the consequences of climate change is one of the most crucial contemporary research topics and it has attracted contributions from every field. However, this has resulted in a superfluity of climate change terminology. Terms are often loosely conceptualized, their definitions may overlap, and their relation to other terms is not clarified (e.g., sensitivity, vulnerability, resilience) [7–9]. In order to circumvent these problems, we attempt to outline a coherent theoretical framework, largely relying on the conceptual approach of the United Nations' Intergovernmental Panel on Climate Change (IPCC) (Figure 1).

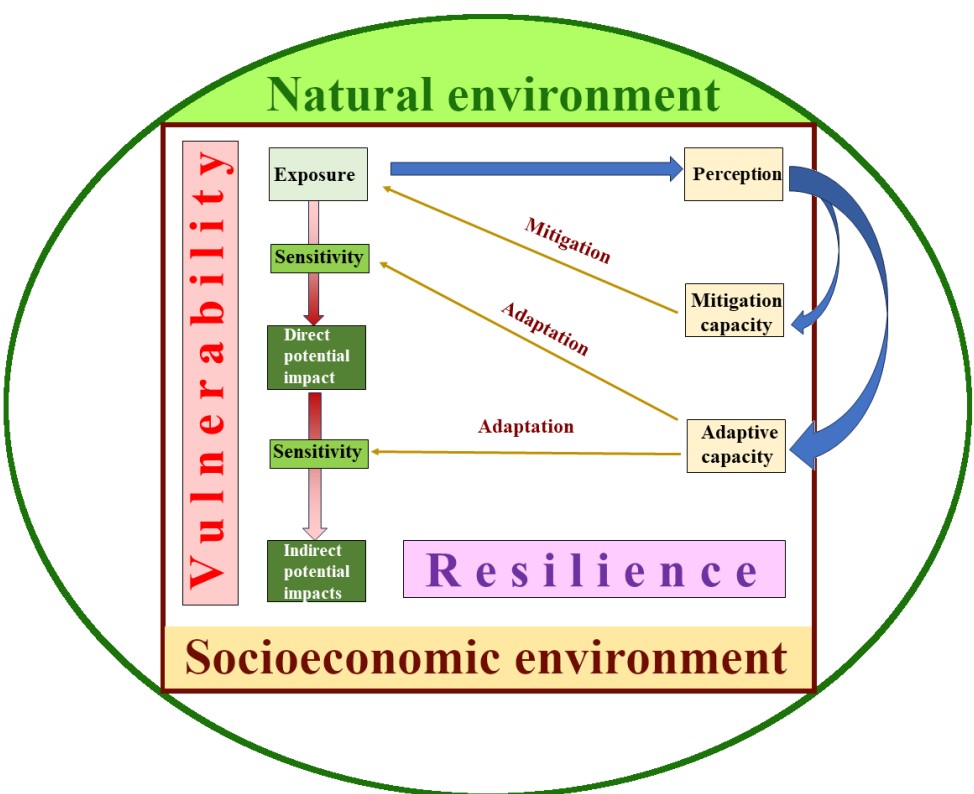

**Figure 1.** Theoretical framework and the relationship between the key terms. Source: own elaboration.

In our study, exposure to climate change is understood as measurable changes in climatic elements, including gradual processes and the frequency of extreme weather events. Our take on climate change exposure includes already occurred, ongoing, and expected changes. Current climatic parameters are also included in our discussion of exposure because they determine the potential impact of expected changes (current stress, threshold crossing). Since the intensity (or even direction) of changes may differ in the case of each climatic element (e.g., precipitation, temperature), exposure can also be disaggregated to components with differing severity.

Exposure to climate change can lead to a wide range of possible environmental, social, and economic impacts. Some of them are direct impacts, while others are indirect, consequent on the direct impacts (e.g., fall in crop production → loss of income → cessation of agricultural activity → reduced ability of the region to provide livelihoods for the

local population). Whether the possible impacts are recognized as potential (or expected) impacts, and the degree of these potential impacts, depends on the sensitivity of the region (social group, system). Sensitivity is a passive, intrinsic quality, derived from the characteristics of the region (social group, system). It may also include environmental, social, and economic components. As with potential impacts, sensitivity consists of multiple layers (e.g., from factors influencing potential crop loss to factors influencing the sensitivity of the local community).

In social sciences, perception is defined as a person's unique apprehension of reality and interpretation of the world. Individual perception influences opinion, judgement, understanding of a situation, and how one responds to it. Thus, with regard to climate change, we use perception not only to explore farmers' recognition of measured climatic changes (gradual processes, frequency of extreme weather events), but also their interpretation of such phenomena. Farmers' perception of climate change may suffer from recall biases and fallacies (e.g., rosy retrospection, frequency illusion). On the other hand, studying perception is key to identifying the elements of climate change crucial for farmers' livelihoods. Interpreting observed changes depends on farmers' knowledge and beliefs as well as personal experience, so we have included these concepts too in the discussion of perception.

Different actions can be taken to offset the negative potential impacts. According to Jones [10], mitigation reduces the rate and magnitude of changing climate hazards. In our study, we define mitigation as conscious acts which aim to reduce exposure to climate change. Mitigation capacity describes the ability of the unit of analysis to conduct acts of mitigation. Acts of mitigation are usually transnational, global endeavors (e.g., abatement of greenhouse gas emissions and greenhouse gas sequestration); at the local and regional levels, mitigation capacity is very limited.

In the usage of the IPCC, adaptation is the adjustment in natural or human systems in response to actual or expected climatic stimuli or their effects, which moderates harm or exploits beneficial opportunities. In line with that approach, we define adaptation as conscious acts which aim to reduce sensitivity to climate change. Since sensitivity can be defined on different levels, adaptation measures may also target different levels (e.g., sensitivity of production, sensitivity of the local community) [11]. Autonomous adaptation typically happens on a smaller, more immediate scale, like individual farms, and involves optimizing production without significant systemic alterations. These adjustments are autonomous because they do not require intervention from other sectors like policy or research. On the other hand, planned adaptations involve more significant, structural changes to tackle the impacts of climate change. These adaptations are more comprehensive, often involving longer timeframes and broader areas, such as entire regions or countries [12].

Research on adaptive capacity came to the fore in 2001 after the third IPCC Assessment Report [13,14]. The most frequently used definition was developed by the IPCC in 2007: "adaptive capacity is the ability or potential of a system to respond successfully to climate variability and change and includes adjustments in both behavior and in resources and technologies" [15].

Vulnerability is the degree to which a system is susceptible to, and unable to cope with, adverse effects of climate change, including climate variability and extremes [15]. Conceptual approaches share the view that vulnerability comprises exposure, sensitivity, and adaptive capacity [16,17]. The CIVAS (Climate Impact and Vulnerability Assessment Scheme) model is an attempt at synthetization and quantification, revolving around these concepts [18,19]. Our paper accepts this established interpretation of vulnerability. Just like potential impacts, sensitivity and adaptation, vulnerability is a multilayered concept. Vulnerability connected to the direct potential impacts of climate change is referred to as outcome vulnerability, while vulnerability connected to indirect potential impact is termed contextual or social vulnerability. This approach recognizes that vulnerability to climate change is embedded in socioeconomic contexts that determine the ability of agents to

cope with external pressures or changes [20,21]. The integration of these two conceptual approaches, as adopted in our paper, is sometimes referred to as the synthetic or hybrid approach to vulnerability assessment [22].

Resilience is another key concept in climate change studies. In their seminal work, Folke et al. (2010:1) use the following definition for resilience in the context of social–ecological systems: "the capacity of a system to absorb disturbance and reorganize while undergoing change so as to still retain essentially the same function, structure and feedbacks, and therefore identity, that is, the capacity to change in order to maintain the same identity" [23]. In our opinion, a unit of analysis can be considered resilient to climate change, if it has the capability to adapt in ways that leave its key characteristics intact while reducing sensitivity to climate change. As discussed above, adaptation measures can address different layers of sensitivity, so resilience too can be interpreted for different layers (e.g., resilience of the local agricultural system versus resilience of the local community). It should also be recognized that "key characteristics" are somewhat idiosyncratic (should crop structure, ownership structure, or farming technology be considered key characteristics?) In our understanding, resilience is a contextual, value-based judgement, implying the persistence of the unit of analysis (or lack of it) in the light of expected impacts and adaptation possibilities.

There is no shortage of existing research that utilized one or more of these concepts to explore the effect of climate change on the agricultural sector. In these studies, many components of climate change exposure have been investigated, including heat [24], frost [25], precipitation [26], and the occurrence of extreme weather events [27]. Some studies indicate the impact of high exposure may manifest in decreasing yields [28] or diminishing crop quality [29–31]. In other cases, it disrupts existing farming practices, for example, causing an increase in heat-related health risks and a decrease in productivity [32]. However, not all impacts described as unfavorable: a study from California found that potential reduction in frost exposure may allow orchard farmers to reallocate their funds to adapt to other impacts of climate change [25]. Some studies also point out that climate change exposure is often coupled with changing market pressures [33,34].

The existing literature explores the environmental aspects, for example, soil texture [35], and social and economic facets [36] of climate change sensitivity alike. One key takeaway from existing research is that the climate sensitivity of certain crops, or even crop varieties, significantly differs [37–39]. Thus, crop structure fundamentally affects the climate sensitivity of agriculture in different regions [40].

Based on preceding research concerning farmers' perception, a large majority of farmers are aware of climate change [41,42]. However, some doubts are also reported [43]. A systematic review found that farmers' perception of temperature change is usually more in alignment with meteorological data than their perception of rainfall [44]. Moreover, some studies found that farmers find it hard to distinguish between climate variability and climate change [45]. Studies also found that socioeconomic factors (e.g., farm size), available information sources, and personal values (e.g., ecocentrism) may all influence farmers' perceptions [46,47]. Research also highlights that farmers' perception is significantly altered by whether they have suffered from extreme weather-related disasters [48].

The scientific literature also takes a marked interest in possible adaptation measures that farmers may take. A systematic review from Asia listed over 30 adaptation practice categories [49]. These possible measures concern crop management [50,51], irrigation and water management [52], farm management [53], financial management [54], physical infrastructure management [55,56], or social activities [57]. As for the factors influencing adaptive measures, another systematic review [58] emphasized the importance of sociodemographic factors [59], income [60], physical capital [61,62], governmental and NGO assistance [63], access to information [64,65], and social networking [66].

Studies of climate change vulnerability and resilience are particularly prone to differing understandings of these concepts [67–69]. Studies with a similar approach to this paper

usually highlight big differences in the vulnerability of territories between or even within countries [36,70,71].

*2.2. Climate Change Challenges of the Horticultural and Grape Sector: Experiences across the Globe*

The horticulture sector, a plant-based branch of agriculture, produces high-value crops like fruits, vegetables, and ornamentals [72]. According to Jaenicke and Virchow [73], the sector plays a vital role in contributing to various Sustainable Development Goals (SDGs) through enhancing food security and health.

With regard to climate change, rising temperatures and changing precipitation patterns are causing abiotic stresses that negatively impact horticultural productivity. These changes can lead to shorter growing periods, reduced water availability, and inadequate vernalization, resulting in decreased yields. Extreme weather conditions, including heavy rain, floods, hail, frost, and droughts, are also challenges to overcome. Global warming brings higher $CO_2$ levels that could potentially increase vegetable yields and their antioxidant content; however, the overall effect of elevated temperatures is likely to be detrimental. Heat stress in vegetables can adversely affect yields and harvest timing due to altered physiological processes like vernalization and winter chilling. In fruit cultivation, climate change exacerbates risks like frost damage and insufficient chilling. The growth rate, premature ripening at higher temperatures, and rising pest populations threaten the quality of special crops, along with shifts in their nutritional profiles, including sugars, acids, and antioxidants. Furthermore, climate change impacts not only plants but also the abundance, diversity, and activity of plant-associated microorganisms, prompting adjustments in the interactions between plants and microbes [30,74–77].

Malhotra [75] highlights the pressing need to enhance the development of horticultural crop varieties that are suited to diverse agro-ecological zones, especially in the face of shifting climate conditions. Unlike annual crops, which can adapt quickly through a variety of cultivars, species, and altered planting times, the establishment and reorganization of orchards demand a more long-term perspective on climate change. As global warming influences the growing environment, the locations and timings for planting certain crops might need adjustment as well.

The horticultural sector covers a wide array of crops, grown in fields, orchards, and under protected conditions such as poly-tunnels and greenhouses [72]. Greenhouse horticulture stands out as a highly intensive agricultural system, aimed at producing high-value goods. It offers advantages like controlled environmental factors (temperature and light), efficient resource use (water, fertilizers, etc.), and advanced technologies (e.g., hydroponics, automation), leading to increased yields, early production, consistent output, and enhanced quality [78]. The advancement of controlled environment agriculture should focus on two main areas, as suggested by Gruda et al. [79]: firstly, high-tech greenhouses with advanced active climate control and management systems; secondly, low-tech greenhouses, employing passive climate control like natural ventilation and screenhouses. Intermediate options are also viable, depending on the crop and regional climate.

The wine sector has been facing similar difficulties due to the vulnerability of vineyards to changing climate conditions [80,81]. The development and growth of grapevines depend on three essential factors: temperature, precipitation, and solar radiation. Thermal conditions heavily influence the physiological development and berry content of grapevines. Moreover, the average temperature is also crucial in determining suitable areas for grapevine farming [82]. Precipitation also has a significant effect on the growth of grapevines, since it influences soil moisture, which is crucial during the stages of planting, budburst, and shoot development [83]. Solar radiation, as the third factor, facilitates the production of sugar, phenolic, and aromatic substances throughout the ripening process, which in turn influences the sensory characteristics of wine, such as taste and fragrance qualities [83]. Any deviation from the typical annual patterns of these factors has an im-

pact on yield and quality, causing adaptations in viticulture operations to accommodate these changes.

However, there is still a lack of knowledge regarding the potential reactions of wine growers to any of these changes in local climates, as well as effective measures for adaptation [84]. Nevertheless, multiple scenarios have been formulated concerning the possible climate change forecasts. According to Schultz and Jones [85], variations in grape composition and wine styles will impact the combinations of different grape varieties and disrupt the traditional pairing of specific types with specific wine areas. In alignment with other studies [86], Hannah et al. [87] forecast a significant decline of 25% to 73% in wine cultivation regions by 2050 in the absence of adequate adaptation measures. In addition, van Leeuwen et al. [88] assert that preserving the quality of wine requires implementing adaptation measures both in the vineyards and cellars, which must involve the use of technological innovation and strategies tailored to specific locations.

Climate change poses environmental and socioeconomic risks and opportunities for wine-producing regions, requiring farmers to adapt and respond accordingly. While adaptation has always been a part of agriculture, the necessity brought about by the present and future effects of climate change is expected to be unprecedented. Hence, it is imperative to comprehend the vulnerability of farming systems to climate change in order to develop effective adaptation measures. Moreover, understanding exposure, sensitivity, and adaptive capacity can provide decision-makers with a framework to prioritize and address climate change issues [84].

## 3. Materials, Methods, and Data

Both qualitative and quantitative methods were utilized to explore the different facets of the interplay between climate change and agricultural activity.

Qualitative methods included field research and semi-structured interviews in four study areas. Interview guidelines and case study schedules were provided before the collection of qualitative data began. The case study and interview schedules used were shaped through a series of discussions within the research team; their aim was to ease the fieldwork and ensure that the empirical research followed similar tracks at each site; they did not constrain researchers' degrees of freedom to adjust their own approach to the specific profiles of field-sites and their own expertise.

Field research began in September 2021. Potential interviewees were approached using the snowballing method. By 1 September 2023, 82 interviews had been conducted at the four research sites.

Semi-structured interviews, most of which had been recorded, were transcribed and prepared for content analysis which grouped the respondents' answers by sub-topic. The interviews provided insights about the farmers' perception of climate change, factors influencing sensitivity, adaptive measures taken, and factors influencing their adaptive capacity.

Quantitative indicators were used for assessing exposure, sensitivity, and factors influencing adaptation capacity. The following steps were taken to prepare the climate change exposure indices:

- Data collection,
- Preprocessing,
- Evaluation of the relation between elevation and climate,
- Interpolation.

Data collection: The source for climate data was the Meteorological Database of the Hungarian Meteorological Service (OMSZ). The EU-DEM digital surface model published by the European Environmental Agency was used to obtain elevation data for Hungary. Finally, the source for the GIS data for the districts was the ArcMagyarország 2021 dataset.

Preprocessing: The structure of the climate data collected required a preprocessing phase. Each observation point has its own Excel table, each containing daily information for their functional period and their geographical coordinates. A Python script was written

to merge the observation points, tailor and filter the information, and create input files for each selected variable.

The following climate indices were selected for interpolation:

- Average temperature,
- Yearly number of heatwave days (with a maximum temperature of over 30 °C),
- Chance of sub-zero (°C) temperature in April or May,
- Average precipitation,
- Yearly number of days with heavy rainfall (over 30 mm),
- Yearly number of days with wind gusts over 17 m/s.

For each index, two input files were prepared, for the time periods 2002–2011 and 2012–2021. Only the 55 observation points containing data for the whole twenty-year period were used in further calculations. The reasons for this short timeframe are threefold:

- While changes in climate are usually examined over a longer period, our results indicate that ongoing climate change in Hungary is rather rapid, causing significant shifts across two decades (see Results).
- Secondly, farmers also make decisions based on a shorter timeframe and may quit or modify their production after a few years of unfavorable weather [89].
- The final practical reason is connected to data availability: available data are much scarcer from the twentieth century, and, for a detailed spatial interpolation, we needed as many data sources as possible.

Evaluation of the relation between elevation and climate: Most climate indices show a high dependency on elevation above sea level. To take this into account during the interpolation, we prepared linear regression models in SPSS to quantify the connection between them, with the climate indices as dependent and elevation as independent variables [90]. The results were significant in each case, except for "chance of sub-zero temperature in April or May". In case of average temperature and heatwave days, $R^2$ was above 0.6, and for precipitation, it was above 0.4, indicating a very strong correlation. Finally, we used the β coefficients obtained to predict the (hypothetical) sea level value of each observational point.

Interpolation: The predicted hypothetical sea level values of the climate indices served as vector data points during the interpolation [91]. The IDW method was used in ArcMap and, based on 55 data points, hypothetical sea level interpolated surfaces were created for each climate indicator. The cell values were altered in Raster Calculator, using the β coefficients from the regression equations and the cell values of the EU-DEM elevation raster. The result is an interpolated surface for the real elevation. The results were also summarized for the area of the four case studies.

Three aspects were used to assess agricultural climate sensitivity to climate change: the drought sensitivity of soils, the sensitivity of crop structure, and the agricultural dependency of local population.

The Hungarian Agrotopographic Database of the Institute for Soil Sciences [92] served as the source of information for drought sensitivity. This GIS database also contains information about the hydraulic properties of the soil. Soils with good water retention and large available water capacity were considered less sensitive. When the index was calculated for Hungarian Districts (including the four case areas) only agricultural areas were taken into consideration. Agricultural areas were delimited using Corine Land Cover (CLC 2018); pastures were excluded.

For determining the crop structure of an area, the classification of the Corine Land Cover (arable land, orchards, vineyards, heterogeneous agricultural areas) as well as the results of the 2020 Agricultural Census were used. The Agricultural Census contains acreage of main arable land crops (wheat, corn, industrial crops, fodder crops). The sensitivity of certain crops is based on their loss ratio provided by the Hungarian Agricultural Risk Management System.

The agricultural dependency of local population is based on the employment share of agriculture, according to the General Population census of 2011.

Finally, Standard Output is used to determine the availability of financial resources, an important factor of adaptive capacity. Standard Output is the average monetary value of the agricultural output from the given area, which is influenced by crop structure, soil fertility, technological investments, accessibility of markets, existing transport infrastructure as well as other factors. The source for the Standard Output was also the Agricultural Census of 2020.

## 4. Study Areas

In order to explore the spatial differences in climate exposure, sensitivity, perception, adaptation vulnerability, and resilience of agriculture, we selected four districts from four different regions of Hungary (Figure 2, Table 1). When the four research sites were selected, our goal was to represent the main geographical (and climatic) divisions of the country (Transdanubia, Great Plain, Northern Hungary). We also paid special attention during the selection process to ensuring that each district displayed different agricultural, environmental, and socioeconomic characteristics. We aimed for diverse case studies to provide a more comprehensive exploration of the factors influencing farmers' perception, adaptation measures, and adaptive capacity.

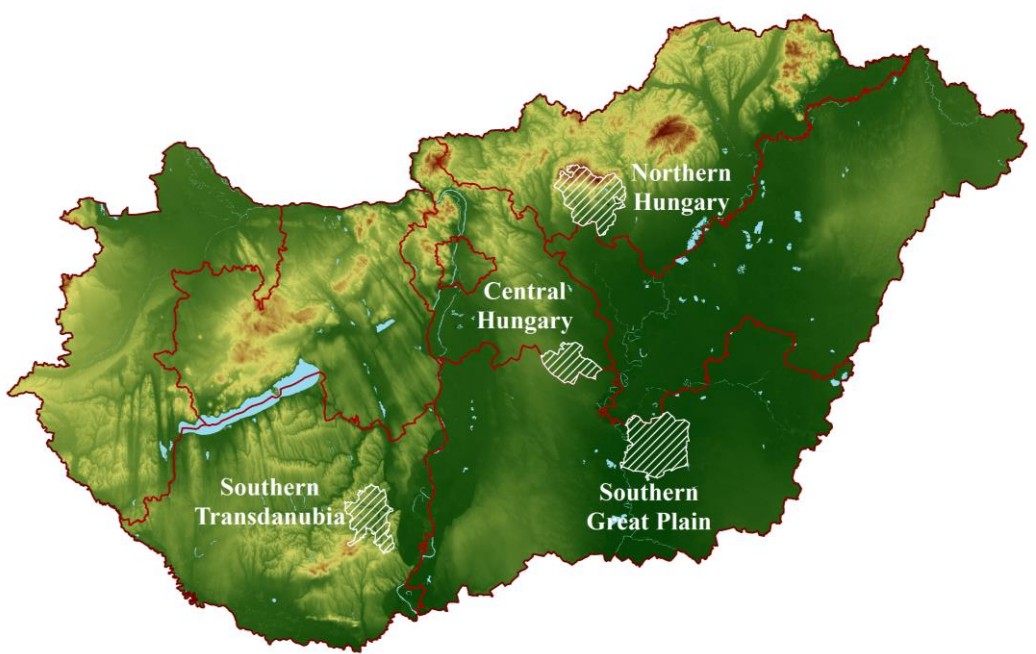

**Figure 2.** The location of the four case districts. Source: own elaboration.

Bonyhád District is located in Southern Transdanubia and characterized by a fragmented, hilly landscape. It is a peripheral, economically stagnating backwater area with a decent small town and several small-scale villages characterized by poor socioeconomic conditions and emerging ethnic segregation. Agricultural employment is in the middle range and farms of mixed profile (arable, fruit, grape, and forest) are common in the area, with an emphasis on fruit production. Small- and medium-scale farms are the most typical farming units.

Gyöngyös District is located in Northern Hungary, on the slopes of the Matra mountains. It is a traditional wine-producing region, with a middle-sized market town surrounded by villages, characterized by sharp socioeconomic stratification. The district has both the highest educational attainment and highest unemployment levels of the case studies, which indicates deep socioeconomic divides. The gravitational pull of Budapest increasingly influences the district as it is slowly incorporated into the ever-expanding

catchment area of the capital. The weight of agriculture in the economy of the district is low; vineyards are still dominated by small-scale plots and large-scale grape farms, though emerging middle-sized farm and processing enterprises have increasingly gained more importance in recent decades [93].

**Table 1.** Main features of research sites.

| Aspects | Southern Transdanubia | Northern Hungary | Central Hungary | Southern Great Plain |
|---|---|---|---|---|
| District name | Bonyhád | Gyöngyös | Nagykőrös | Szentes |
| Location, physical geographical characteristics | Rolling hills of Tolna, fragmented landscape | Foothill of Mátra Mountains | Flat sandy soils of Great Hungarian Plain | Alluvial soils of Great Hungarian Plain |
| Size, population, settlement network | 476 km$^2$, 28,000 inhabitants in 25 municipalities, seat is a small town of 12,000 | 751 km$^2$, 69,000 inhabitants in 25 municipalities, seat is a middle-sized market town of 28,000 | 349 km$^2$, 27,000 inhabitants in 3 municipalities, seat is market town of 23,000 | 814 km$^2$, 37,000 inhabitants in 8 municipalities, seat is a middle-sized market town of 25,000 |
| Complex profile: spatial, economic, social | Internal periphery of Transdanubia with a medium-developed small town center (Bonyhád) and tiny villages undergoing segregation, lowest level of education | Diverse economy (tourism, service sector, FDI investments) with a strong medium-sized central town, within Budapest's commuting zone, socioeconomically divided | Declining Great Plain agrarian town (Nagykőrös) and surrounding villages, in the Functional Urban Area of Kecskemét, slower demographic erosion | Internal periphery of southern Great Plain centered around medium-developed agrarian town (Szentes), lowest rate of unemployment |
| Agricultural profile | Orchards Consolidated medium farms, stable small farms | Grape and wine production Emerging medium farms, but small and large businesses still dominant | Mixed Consolidated medium businesses | Vegetables Consolidated medium farms, stable small farms |
| Number of interviews | 21 | 23 | 23 | 15 |

Nagykőrös District is located in Central Hungary, the Danube–Tisza Interfluve territory of the Great Hungarian Plain. Land of differing qualities covers the area, ranging from fertile fields appropriate for arable farming to poor quality, sandy soils suitable only for either labor-intensive agricultural production, notably wine and fruit production, or extensive types of cultivation, such as forest or grass. The center of the district is a middle-sized but declining market town, with two satellite villages. The district falls into the catchment areas of both the capital city (Budapest) and the neighboring regional center (Kecskemét).

Szentes District is located in the Southern Great Plain, in the alluvial plain of the Tisza River. Its fertile fluvisols give rise to a diverse agriculture, including horticulture with glasshouses and poly-tunnels. While Szentes is a potent middle-sized town, the district falls outside current axes of development. The weight of agriculture in the local economy is high due to the dominance of intensive horticultural farms with a vegetable growing profile, which are increasingly run by middle-scale enterprises.

To facilitate differentiation, we will usually refer to the case studies by the name of their encompassing region.

## 5. Results

### *5.1. Exposure*

Since the beginning of the new millennium, three out of the six selected indices underwent a significant unfavorable change at the national level. As Figure 3 indicates, there has been a rapid increase in the average annual temperature (over 0.75) in Hungary over the last two decades. This was accompanied by a drastic rise in the number of heatwave days (days with a measured maximum temperature of over 30 °C). Despite this, the chances of late spring sub-zero temperatures also increased during the period under examination. The problem is further exacerbated by the fact that, due to warmer days in February and March, the vegetation period for crops started earlier than decades before. Other (local and global) findings also underline this issue [94,95].

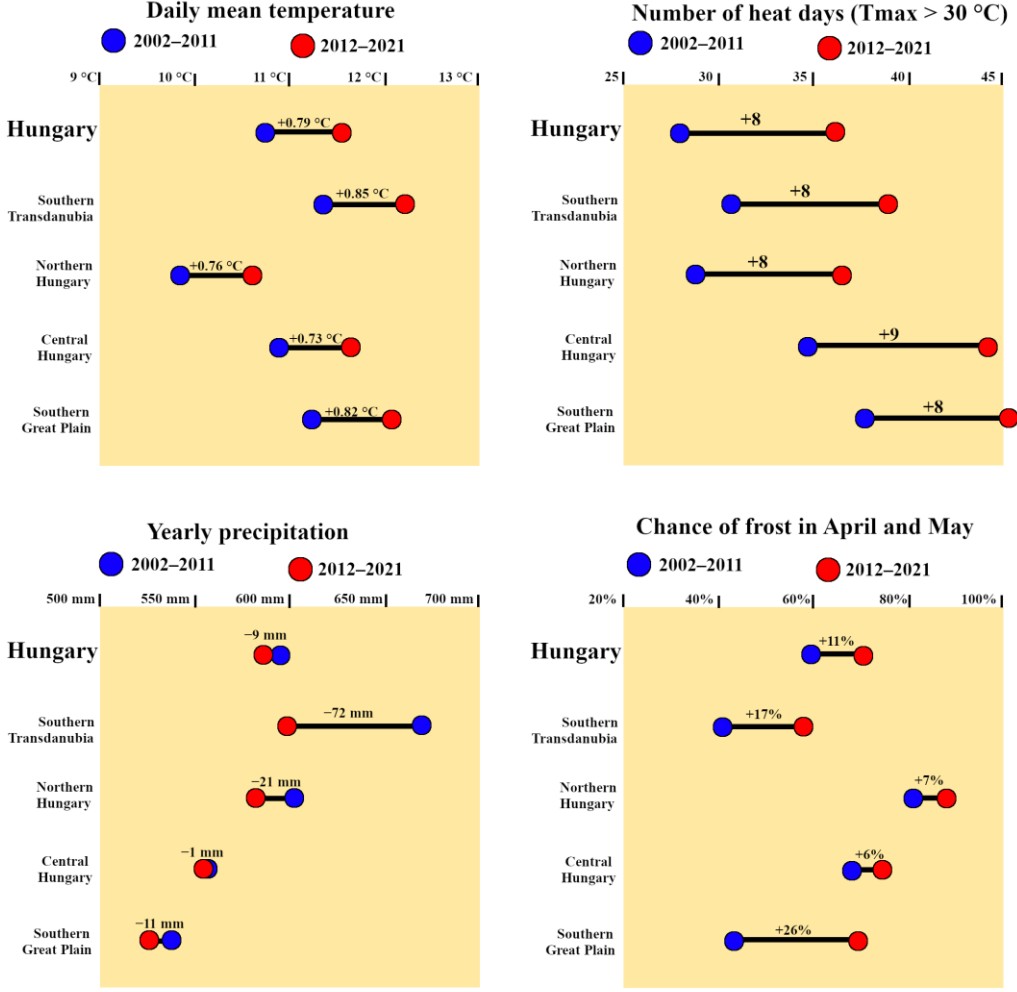

**Figure 3.** Changes in climate parameters in Hungary and in the four case areas in the last two decades. Source: Own calculation based on data from the Meteorological Database of the Hungarian Meteorological Service.

On the other hand, at national level, the average annual rainfall was only slightly reduced. However, in the case of precipitation, the country average masks significant spatial differences. With regard to both the number of days with heavy rainfall and the frequency of stormy days, we observed no significant changes at the national level in Hungary. Furthermore, clear spatial trends were not discernible. This emphasizes the challenges and uncertainty associated with detecting and forecasting extreme weather events.

Spatial analysis of the selected indices paints a more nuanced picture (Figure 4). Both average temperature and the number of heatwave days are at their highest in the Southern

Great Plain, while they are at their lowest in mountainous areas and the western part of the country. Changes in temperature and heatwave days, while universal, are somewhat more pronounced at the edge of the Great Hungarian Plain. The frequent occurrence of sub-zero temperatures in April or May is influenced by multiple local factors, such as soil characteristics and relief: for example, the Danube–Tisza Interfluve (where the Central Hungarian research site is located) is more prone to late spring frosts than its surroundings. The southwestern part of the country, including Southern Transdanubia and the mountainous areas, experience the highest rainfall, while central and southeastern parts of the Great Hungarian Plain receive the smallest amount of precipitation. Change in precipitation, however, shows a very distinct differentiation. In the western and especially the southwestern parts of the country, we found an increase in precipitation. On the other hand, significant reductions were observable in some parts of Southern Transdanubia (including Bonyhád district), and the northeastern part of the country. This is in line with the aforementioned European climate projections.

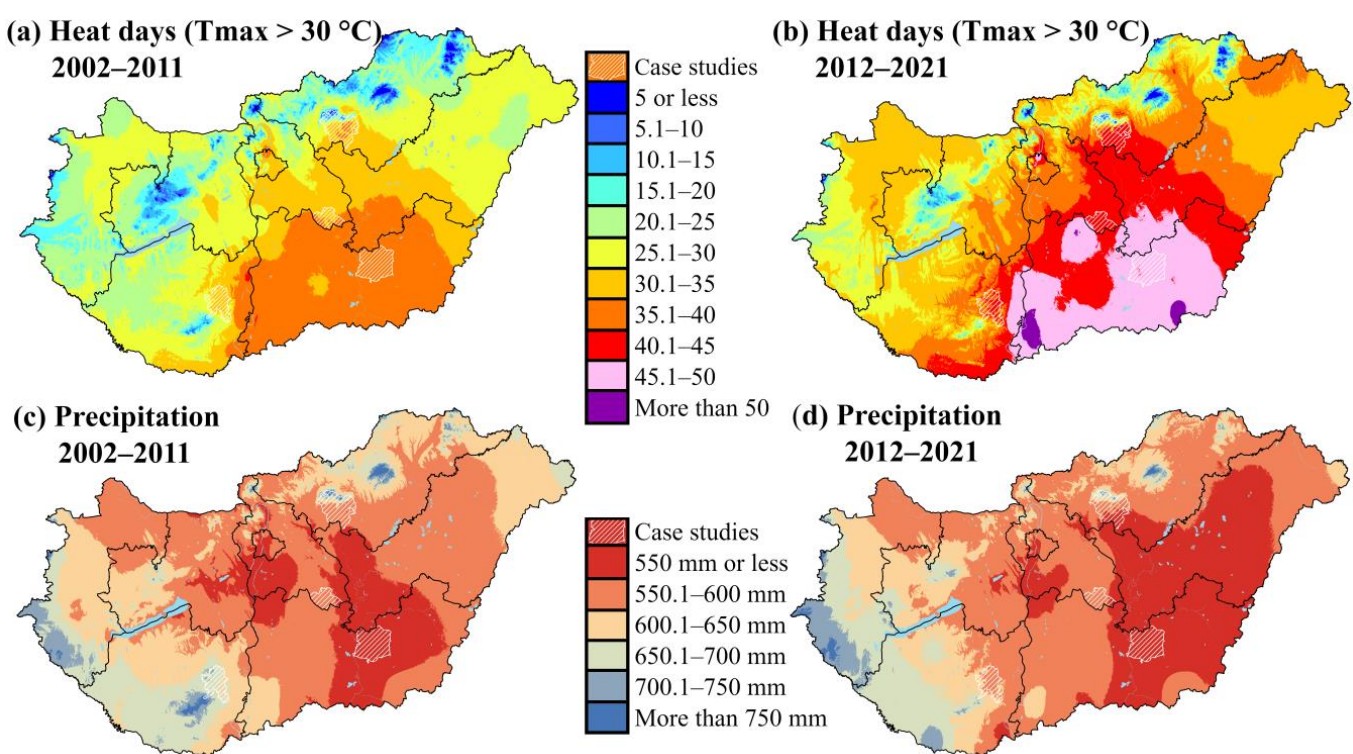

**Figure 4.** Yearly average number of heatwave days (days with a maximum temperature of over 30 °C), 2002–2011 (**a**), 2012–2021 (**b**); and yearly average precipitation 2002–2011 (**c**) and 2012–2021 (**d**). Source: Own calculation based on data from the Meteorological Database of the Hungarian Meteorological Service.

Considering the four case areas, Southern Transdanubia can be described with moderate exposure to climate change. While its position is still more favorable in terms of heat days, precipitation, and chance of late spring frost, this district also experienced the most radical decrease in precipitation.

Northern Hungary is characterized with moderate exposure to climate change. Due to the Matra mountains, the mean temperature and the number of heat days are average. However, the chance for late spring frosts is exceptionally high.

Both Central Hungary and the Southern Great Plain can be described with high exposure to climate change, with an extremely high number of heat days, and moderate chance for late spring frosts. While the precipitation did not decrease significantly in either district, it is still very low, and these areas often suffer from droughts.

*5.2. Sensitivity*

Sensitivity to climate change may include environmental, social and also economic components. One environmental factor is the drought sensitivity of the soils. Based on the hydraulic properties of the soils, our index implied drastic differences between the case studies. If we compare them to the values of all Hungarian districts (175), the sandy soils of Central Hungary fall in the first (worst) quintile, while Southern Transdanubia, with its brown earth, is in the fifth (best) quintile.

> *"We farm on poor soil, valued at only one golden crown. The consequence of climate change is that you cannot grow anything on it. In the past there was so much water that the ditches were full of leeches. Now the water is 4–6 m down." (Central Hungary)*

According to the data from the Hungarian Agricultural Risk Management System, the sensitivity of different crops to climate change, indicated by the loss ratio, shows a marked differentiation. Based on the years 2018–2021, the loss ratio (the ratio of payments and subsidies) is the lowest for cereals (wheat~1), average for industrial crops (rapeseed~3.5), and extremely high for certain fruit species (apricot~36). There are great differences even between the varieties of the same species:

> *"I asked the National Agrarian Innovation Centre to give me old, hundred-year-old, varieties to see what they were capable of now. I am a curious person; I am interested in what my grandfather or his forebears could produce here. Well, for one thing, they produce very little, for another they don't like to be in shaded poly-tunnels. They cannot cope with the climate. So, these varieties have been resting in the gene bank for a hundred years, only the climate is not the same as it was then." (Southern Great Plain)*

This makes crop structure one of the most crucial components for assessing the sensitivity of local agriculture. According to the data from Corine Land Cover and the Agricultural Census of 2020, Central Hungary and Northern Hungary are both in the second quintile (worse than average), while Southern Transdanubia falls in the third quintile (average), and Southern Great Plain is in the fifth quintile (best). Some interviewees from the Southern Great Plain felt similarly and noted that their crops are largely protected from the elements under the poly-tunnels and glasshouses. However, this draws attention to the technological component of sensitivity, which, as a consequence, is the most pressing in the Southern Great Plain:

> *"We grow under cover, you know, so for us, if there is an issue, the problem will not be with our plants but with the whole installation. And in this respect the weather can be very hectic. There are times when it is absolutely fine and others when we are simply incapable of protecting the installation itself." (Southern Great Plain)*

As pointed out in the previous sections of this paper, climate sensitivity, as well as agricultural climate sensitivity, consists of several layers, related to either direct or indirect potential impacts. The agricultural dependency of local communities describes agricultural sensitivity in the largest sense: if the local agriculture becomes unfeasible due to the direct potential impacts, to what degree will it harm the livelihood of the population (indirect impact)? According to the data from the General Population Census of 2011, agricultural dependency is highest in the Southern Great Plain (first quintile), worse than average in the Southern Transdanubian and Central Hungarian case districts (second quintile), and better than average in Northern Hungary (fourth quintile). This result also indicates that opportunities for economic diversification as part of a dynamic agglomeration reduce the agricultural climate sensitivity of an area, while isolated, peripheral areas tend to be more agro-dependent.

In summary, in comparison with other districts in Hungary, the sensitivity scales of the four study areas are as follows:

The agricultural climate sensitivity of Southern Transdanubia is moderate, with low soil sensitivity, average crop sensitivity, and higher than average agricultural dependency.

The climate sensitivity of the Northern Hungarian case study is also moderate, with lower than average soil sensitivity and agricultural dependency, and higher than average crop sensitivity.

The climate sensitivity of Central Hungary is high, with higher than average crop sensitivity and agricultural dependency, and high soil sensitivity.

Finally, the sensitivity of the Southern Hungarian Plain is moderately high, with low crop sensitivity but high technological sensitivity, higher than average soil sensitivity, and high agricultural dependency.

*5.3. Perception*

As discussed above in the theoretical section, our conceptualization of climate change perception includes farmers' recognition of measured climatic changes (gradual processes, frequency of extreme weather events), and their interpretations based on their own experience, knowledge, and beliefs.

In each study area, most interviewees reported an increasing frequency and severity of extreme weather-related events in the last five years. Most of them also agreed that these occurrences were caused by climate change; they therefore expect their repeated long-term recurrence:

> *"It is indeed climate change and I fear it a lot. At all events, we must prepare for the fact that there will be ever more serious crises [caused by climate change]." (Southern Transdanubia, apricot farm owner)*

> *"Well, I don't think anything is going to stop global warming, because I think it's going to get even worse. (Northern Hungary, viticulture farmer)*

However, we also encountered a few cautious, doubtful opinions, such as a farmer expressing cautious skepticism:

> *"My view on climate change itself is that I do not say that I do not believe in it, but I think that I am not able to judge what its effects on us are, or that it was not the case 100 years ago. When my granddad was alive, he said that in nineteen fifty something (…) they could only harvest at night because it was so hot." (Southern Transdanubia, owner of a mixed fruit plantation)*

The interviewees mentioned a wide range of different climate phenomena which negatively affected their production, including shorter winters, summer heatwaves, increased UV radiation, late spring frosts, lack of water, an increasing frequency of intense storms, and strong winds. The latter are especially problematic since they can cause significant damage to poly-tunnels.

In the Central Hungary research area, one farmer, who has been working in horticulture since childhood, said:

> *"Twenty-five years ago, there were spring frosts too of course, as I said, but not like this: a $-15$-degree morning frost which decimates apricots so that by the time we get to the first flower opening, a half of the tree, the bud is dead" (Central Hungary)*

> *"There have always been frosts, of course, but not three years in a row, there was never that, and now it is the fourth year" (Central Hungary)*

> *"(…) We had this storm again, I don't know, it took away 16–18 of our poly-tunnels … I don't remember winds like that when I was a kid." (Central Hungary)*

The interviewees also brought up examples of indirect damage that can be traced back to the consequences of changing climatic conditions:

> *"They (common cutworms) simply lay waste to the fruit out on the sand. Literally, lay it waste. In truth it is not a new phenomenon, but in terms of size, there is now an invasion, and I don't know what causes it. Probably climate change. Because in the past there was*

*a season when the ground was frozen. Not anymore. They are often there in the winter too, eating the crops."* (Central Hungary)

*"On Mud Hill, there used to be water sources that used to give water all through the year. These have all now disappeared . . . The poor animals are thirsty. They cannot find water up in the forest. What else can they do? They come down amongst the vines to graze."* (Northern Hungary)

We identified the following factors that may influence the farmer's perception of climate change:

Age of the farmer: those who started farming more than a decade ago and have their own experience are likely to support the majority view and accept climate change as a reality.

Relative unpredictability of climatic occurrences across locations: occurrences of heavy storms, hail, or frost can hit extended territories but sometimes they are very localized and hit a couple of the settlements in a district, whilst others escape.

*"(. . .) We have been growing apricots in K since 2002. Only one crop has been spoiled by hail since then. By contrast, we established a plantation in M in 2010 which has been producing since 2013. Over the nine years since then, the crop has been spoiled by hail four times, and the plantations are ten kilometers apart. I don't know how much hail there was in M in the 1990s."* (Southern Transdanubia, young farmer)

The complexity of climate change: As noted above, negative effects often emerge indirectly. Farmers' perceptions can be affected by whether or not they realize the intricate web of climate-related changes and connect seemingly independent observations.

The perceptions of the interviewees also differed as a consequence of the different characteristics of the study areas.

In Southern Transdanubia, the emphasis was put on severe late spring frosts, with a significant loss of apricots (90%) experienced in three of the last four years. The interviewees also mentioned drought in the summer as a recurring problem, and the appearance of new varieties of pests.

In Northern Hungary, the observed negative changes included drought, extreme heat in the summer, heat stress, and the emergence of new pests and diseases. Damage from wild animals was also mentioned.

In Central Hungary, spring frost damage was also brought up, along with extremely warm and dry weather in summer, heat stress, and a fall in the groundwater level.

In the Southern Hungarian Plain, the emphasis was put on more frequent strong winds and storms which can damage poly-tunnels. They also mentioned the earlier start of the vegetation period, which increases the potential damage from late spring frosts, and unbearable summer heatwaves. The interviewees also complained about new invasive pests, e.g., new types of shieldbugs, that can overwinter in the absence of serious winter frosts.

### 5.4. Mitigation and Adaptation

As noted above, mitigation efforts—acts which aim to reduce exposure to climate change—are mostly significant on a transnational or global scale. However, there is one instance where local or regional mitigation can make a difference: hail suppression. Hungary operates a dense network of hail suppression stations (incorrectly referred to as 'ice cannons'). These insert silver iodide into the atmosphere to hinder the formation of larger hailstones. The results suggest that the interviewees do indeed feel the difference:

*"In this region, there are many places where they protect the orchards from hailstones by 'firing cannons' over them. In the past there used to be tremendous hailstorms. Touch wood, there haven't been any that bad in the last couple of years."* (Southern Great Plain)

We encountered a larger number of adaptation measures. As discussed in the Theoretical framework section, adaptation—acts which aim to reduce sensitivity to climate

change—may address different layers of sensitivity, related to direct or indirect potential impacts.

Some adaptation measures can reduce vulnerability while leaving the current agricultural practices mostly intact. These include shifts in the timing of labor activity. With earlier planning, glasshouse farmers readjusted themselves to the shorter winters. Vine growers may also start their pruning earlier. To avoid unbearable heatwaves, farmers schedule work for the early morning, or even at night:

> *"And that is when a form of night work became common, because the workers could not cope with the heat, so they work amongst the plants at night wearing head torches." (Southern Great Plain)*

High-tech investments may also come into play in the battle against heat, especially for the glasshouse farmers of the Southern Great Plain:

> *"The worse the quality of the installation, the more seriously it is affected by climate change. Plants also survive better in more spacious ones . . . In these modern installations, the climate is better inside than out." (Southern Great Plain)*

However, some also pointed out the usefulness of more affordable, low-tech solutions. For example, applying white paint to the poly-tunnels can successfully reduce extreme warming and UV radiation. The downside is that it cannot be regulated, unlike some high-tech solutions. Other low-tech (but labor-intensive) investments include establishing water reservoirs for irrigation; these are the most common defense tools against drought. Some farmers installed ice-nets against hailstorms, while against the critically damaging spring frosts, all solutions are welcomed (ventilators, anti-frost candles, fumigation, frost protection irrigation, spraying biostimulators, fog spraying, wind machines).

Another direction farmers may take is to find nature-based solutions for their plight. In addition to irrigation, farmers also reported practices to improve soil structure and water balance, such as no-tillage, organic fertilization and mulching, and the incorporation of straw residues.

> *"We have been working without tillage for quite a long time. Organic fertilization specifically! . . . this is also good for improving soil structure because the organic fertilizer loosens the soil." "All we do is to have mulch covering for winter, for example . . . That's why we have rye silage, for example, which also prevents erosion in winter." ". . . we never bale rye or straw. We always put it in (the soil). It retains water at least and creates a little humus." (Central Hungary)*

Other adaptation measures lead to significant changes in current agricultural practice, for example, altering the crop structure. Fruit farmers from Central Hungary and Southern Transdanubia alike reported that when replacing and renewing plantations, they choose varieties and species with later flowering to reduce the damage caused by spring frosts.

> *"I planted the latest varieties of plums and apricots. Plums are better than sour cherries in terms of climate change." (Old-aged fruit grower, Central Hungary)*

In Southern Transdanubia, frost-resistant apricot saplings are imported from abroad (mostly Italy). Southern Transdanubian farmers also try to diversify their crop structure by planting new fruits such as elderberries and figs. In the greenhouse horticultures of the Central Hungarian research site, a structural change can also be observed: vegetable growing is increasingly being replaced by fruit growing, which means not only strawberry growing, but also planting fruit trees in poly-tunnels on dwarf rootstocks to protect them from spring frosts.

We found that adaptation measures against climate change, often facilitated by other factors such as changing consumer preferences and labor shortage, also influenced the adaptation choices that farmers make [96]. For example, it is easier to find manual labor for fruit growing, as it is conducive to remuneration by piece rates, which workers prefer. Market needs and consumer habits also encourage fruit rather than vegetable production.

*". . . we're giving up vegetables. Well, the way I see it, my kids and this generation, they're eating less and less vegetables . . . the world is changing and we can sell fruit better."* *(Young greenhouse horticulturist, Central Hungary)*

One interviewee reported planting a variety of plum that stays on the tree for a long time, and does not drop off, so that they can prolong the harvest. These findings underscore the necessity of a social vulnerability approach when dealing with the consequences of climate change.

Finally, some adaptation measures are associated with shifting away from agricultural activity. This can be of a less radical nature, such as diversifying toward tourism:

*"I would like to build a guesthouse now. I would create a Mediterranean garden here. I want to stand on two feet. At harvest time, make a link with wine-tasting to get around the labor shortage. People will come from the towns, they will come and happily pick fruit, drinking wine throughout the day, a little lunch, dinner. I see the future in agrotourism. Like in Italy and France. It works really well there; in my view, it could work here too. I will create a little terrace here, with places to sit out, and I'll get some palm trees." (Young farmer from Southern Transdanubia)*

In other cases, the cessation of agricultural activity is final; in the Central Hungarian case area, some old orchards are no longer being replaced by new ones but by forests. The share of vineyards has decreased significantly in recent decades. This is also connected to the labor shortage problems of small farms based on manual labor; crop production on arable land using contractual mechanical labor persists:

*"One of the things to know about Nyársapát is that 30 years ago, the village had more than 300 hectares of vineyards. Today, you could say that there are zero . . . I don't see 300 hectares of vines again in the village's future, and there won't be any orchards either. Only forest. There will be field crops, rye, and the like. Rye will cover the costs: that's what it's for." (Old-aged fruit grower, Central Hungary)*

We identified the following factors, which influence adaptive capacity:

Age of the farmer: we found that those over 60 find certain decisions harder to make, especially if succession of the farm is uncertain or not possible. Age is related to time perspectives, and such partially related factors as attachment to farming and whether young people see career opportunities in farming.

*"(. . .) When my dad said it, he was 59 and he felt that he did not want to be too active in fruit growing; he wanted to sell the land, but I was at school still at the time and said, absolutely not, I will farm it. He thought it was a lot of work and did not see much future in it, which I and the other farmers did; he did not even like growing apricots much, but I do, a lot." (Young farmer, Southern Transdanubian site)*

Self-organization, networking, and knowledge: we found that the Southern Transdanubia case study revealed the highest degree of dynamism in agriculture, a consequence of the new opportunities offered for fruit farming. This opportunity was exploited by a group of farmers of diverse social backgrounds and local attachment who collaborated rather on the basis of common interest. They are active members of a regional Producer Organization (PO) and most of them are founding members of a locally based co-operative aimed at joint marketing. They use the sales services of the co-operative in a flexible manner, whilst they regularly draw profitably on the PO's advisory capacities. Informal networks operate equally intensively as well as forms of formal co-operation: sharing experiences and learning from one another is a common feature of their community, and these contribute significantly to their success.

*"There is one useful thing to do against climate change: travel to places that have already been struggling with it for years. So, tomorrow I am going to Italy to see Italian growers with T. Like a time-traveler (. . .) to import foreign technologies. New sprays, new breeds, ice-net systems, but mainly knowledge, information." (. . .) There were study trips to*

*Italy, Austria, and other countries more advanced than us. It was lucky for us that we did not have to be geniuses; we just had to bring things home." (Middle-aged fruit farmer)*

Availability of financial resources: most of the adaptation methods mentioned above require investment. So, in order for the farmers to take these adaptation measures, it is crucial for them to have their own financial resources available as well as the expectation of an eventual return on their investment. By contrast, in the absence of accessible financial resources or subsidies, a radical transformation of production structures is also unlikely. We used the Standard Output index, which indicates the average monetary value of agricultural output at a given location, to measure the income-generating capacity of the case study areas. The formulation of the index is described in the Methodology section. There are a total of 175 districts in Hungary. The Southern Transdanubia case and the Northern Hungary case are in the third quintile (average income-generating capacity) while the Central Hungary and Southern Great Plain cases are in the fifth quintile (best income-generating capacity). This also underscores that funds available can used for different things, for expensive adaptation measures involving technological investments (Southern Great Plain), but also facilitating radical transformation (Central Hungary).

The different characteristics of the study areas also left their marks on adaptive capacity and preferred adaptation measures.

The adaptive capacity of the case study in Southern Transdanubia is high, due to the presence of a group of mostly middle-aged farmers, who are involved in strong networks, are active members of a Producer Organization (PO), and unceasingly expand their knowledge and look for opportunities to adapt to climate change. These adaptation practices include traditional defensive measures against spring frosts (e.g., heating, frost protecting irrigation), looking for and importing frost-resistant varieties from foreign countries (mainly from Italy), and diversifying their income by branching out in the direction of rural tourism.

The adaptive capacity of Northern Hungary is moderate. In this case study, we found that reluctance to engage in networking (mostly within villages) reduced the social capital necessary for adaptation. The owners of the largest farms are the most open to innovation. The costs of constructing irrigation infrastructures to reduce drought sensitivity are so high as to be unfeasible because of the district's hillside location. Diversification possibilities between crop varieties are limited. Diversification to wine tourism is possible in theory, but there are already strong contenders (Eger) in the region. Adaptation measures include thwarting sunburn, hail damage, and damage by wild animals. Growing uncertainty leads to a further unfavorable adaptation measure: some farmers are avoiding long-term investments.

The adaptive capacity of the Central Hungarian research site is low. There, we found an absence of self-organization and networking, and low general social capital. The absence of networking hinders the spread of possible good practices, and high-tech and low-tech solutions. In farms run by ageing farmers, both fruit production and grape production are in decline, leaving little room for crop diversification. Apart from a few innovative attempts, such as planting dwarf fruit trees under poly-tunnels, adaptation measures are mostly reactive (e.g., spring frosts, hail damage). We also encountered farmers who are giving up agriculture in favor of afforestation.

The adaptive capacity of the case study on the Southern Hungarian Plain is high. Here, we encountered examples of technology-intensive adaptive measures: some farmers investing in cutting-edge technology to keep the inner environment in glasshouses under better control. Low-tech solutions, like applying paint to poly-tunnels, were also successful. Some farmers experimented with innovative diversifications of crop structure. The high levels of institutionalized social capital among producers (members of Hungary's largest Producer Organization) have been promoted since the early 2000s, facilitating the sharing of ideas and dissemination of good practice.

## 6. Discussion and Conclusions

Both the quantitative analysis and the interviews uncovered intriguing differences between the four case areas (Table 2). The key takeaways are the following:

- Despite the relatively small geographical extent of Hungary, there were significant differences between the climate exposure and sensitivity of the case studies, resulting from differing macroregional climate change trends and their diverse physical geographical conditions. This finding underscores the importance of locality-based analysis in climate change research.
- The interviewees almost unequivocally agreed that climate change is detectable and disrupts agricultural activity. They evaluated the importance of perceived changes based on their own agricultural profile. In the case of extreme weather events, there was a discrepancy between exposure and perception. This can be explained by high local differences in occurrence, which the interviewees also commented on, as well as by limitations on their powers of recollection.
- The interviewees listed a wide range of adaptation practices. Some of them were applied universally in all case areas (e.g., spring frost protection), while others required a greater propensity for experimentation and innovation on the part of the farmers.
- High local social capital and strong networking facilitated the dissemination of good practice. We found that the role of active producer organizations was crucial, especially if they were complemented by informal, face-to-face networking.
- Our results indicate that neither high climate exposure, nor unfavorable socioeconomic conditions exclude the possibility of successful adaptation. The case area in Southern Transdanubia was characterized by poor socioeconomic conditions while the Southern Great Plain could be described as suffering from high exposure. Our results nevertheless indicate that the adaptive capacity in these regions was high: farmers were experimenting with, and actively looking for, innovative practices.
- Furthermore, our results indicate that the partial isolation of a micro-region may even provide the incentive for adaptation: the relative lack of opportunities makes abandoning an important leg of the local economy less plausible. On the other hand, for regions integrated into dynamic urban areas, it is easier to give up on agriculture.
- However, we hypothesize that there are limits to this effect. An economically struggling outer periphery with an impoverished and deprived population may lack even the rudimentary skills for adaptation.

As discussed in our theoretical framework, vulnerability comprises exposure, sensitivity, and adaptive capacity. Table 2 presents notable differences between the components for the case studies:

- Southern Transdanubia can be described as having the lowest vulnerability of the four case studies, with a moderate exposure and sensitivity, and high adaptive capacity.
- Northern Hungary can be labeled with moderate vulnerability. As in Southern Transdanubia, this case study has moderate exposure and sensitivity, but also moderate adaptive capacity.
- Central Hungary has the highest vulnerability of the four case studies, with a high exposure and sensitivity, but low adaptive capacity.
- Finally, the case study on the Southern Great Plain can be described as having moderately high vulnerability. While the exposure of the district is high and its sensitivity is moderately high, the high adaptation capacity of the district decreases its vulnerability.

Following the hybrid approach to vulnerability, these evaluations integrate both the concept of outcome and contextual vulnerability. In case areas with higher climate exposure, outcome vulnerability poses the higher risk, while in the case of areas with low adaptive capacity, social vulnerability is accentuated.

**Table 2.** Main findings of research sites.

| Aspects | Southern Transdanubia | Northern Hungary | Central Hungary | Southern Great Plain |
|---|---|---|---|---|
| Climate change exposure | Moderate | Moderate | High | High |
| Climate change sensitivity | Moderate (soil: low; crop structure average; agro-dependency: high) | Moderate (soil: low; crop structure high; agro-dependency: low) | High (soil: high; crop structure high; agro-dependency: high) | Moderately high (soil: high; crop structure high; technology: high; agro-dependency: high) |
| Climate change perception | Severe frost in the last three years, 90% of apricots lost, drought during the summer and new varieties of pests | Drought, extreme heat in the summer, heat stress; emergence of new pests and diseases, Damage by wild animals | Repeated frost damage, extremely warm and dry weather in summer, heat stress, low level of groundwater | Frequent strong winds and storms, late spring frosts, unbearable heatwaves, new types of pests (e.g., shieldbugs) |
| Adaptive capacity | High strong networks, active Producer Organization, active collection of know-how in foreign countries | Moderate Networks mostly within villages, only larger farms are open to innovation | Low absence of self-organization, networking, lack of social capital | High strong networks, active Producer Organization, knowledge transfer |
| Adaptive practices | Fight against frost: heating, frost-protecting irrigation, ventilation, change in breed toward less frost-sensitive species Irrigation against drought Ice-net against frost, diversification of on-farm and off-farm activities (tourism) | Changed timing of pruning, reducing exposed leaf area, reducing planting density; protecting berries from excessive sunlight, soil management, mechanization, diversification toward fruit plantations and tourism | Variety change, restructuring toward afforestation, frost protecting irrigation, ice-net, implementation of irrigation | investing in cutting-edge technology in glasshouses, low-tech solutions—applying paint in poly-tunnels, rescheduling labor activity to adjust to changing seasons and avoid heatwaves |

As discussed above in detail, adaptation measures range from subtle changes, which leave the current agricultural practices intact, to radical alterations of existing production structures or even land use patterns. In line with our definition of resilience, agricultural systems, which have the capacity to enact change, which preserve the key characteristics of the system and are also capable of adapting it, can be considered resilient to climate change (Figure 5). Our analysis indicates that the agricultural systems of the Southern Transdanubia, Northern Hungary, and Southern Great Plain case studies can be considered resilient. The case district in Central Hungary, however, is inclined to drastic adaptation measures (establishing commercial forests in place of orchards). In the light of this, we consider the agricultural system of Central Hungary non-resilient to climate change.

The findings of this paper generally align with the conclusions of previous research (refer to Section 2), albeit with some notable nuances. While some studies suggest favorable changes in frost exposure, our analysis predicts an increasing occurrence of late spring frosts. The difference may lie in the varying preparedness of farming systems. In agricultural regions such as California, where farmers are well-equipped to prevent frost damage, the reduction in the absolute number of days with sub-zero temperatures may decrease the costs of expensive safeguard measures. However, in our case study areas, where most farmers are ill-prepared to thwart frost damage, the critical factor is the occurrence or absence of harsh frosts during the vegetation period, not merely a reduction in the absolute number of sub-zero days.

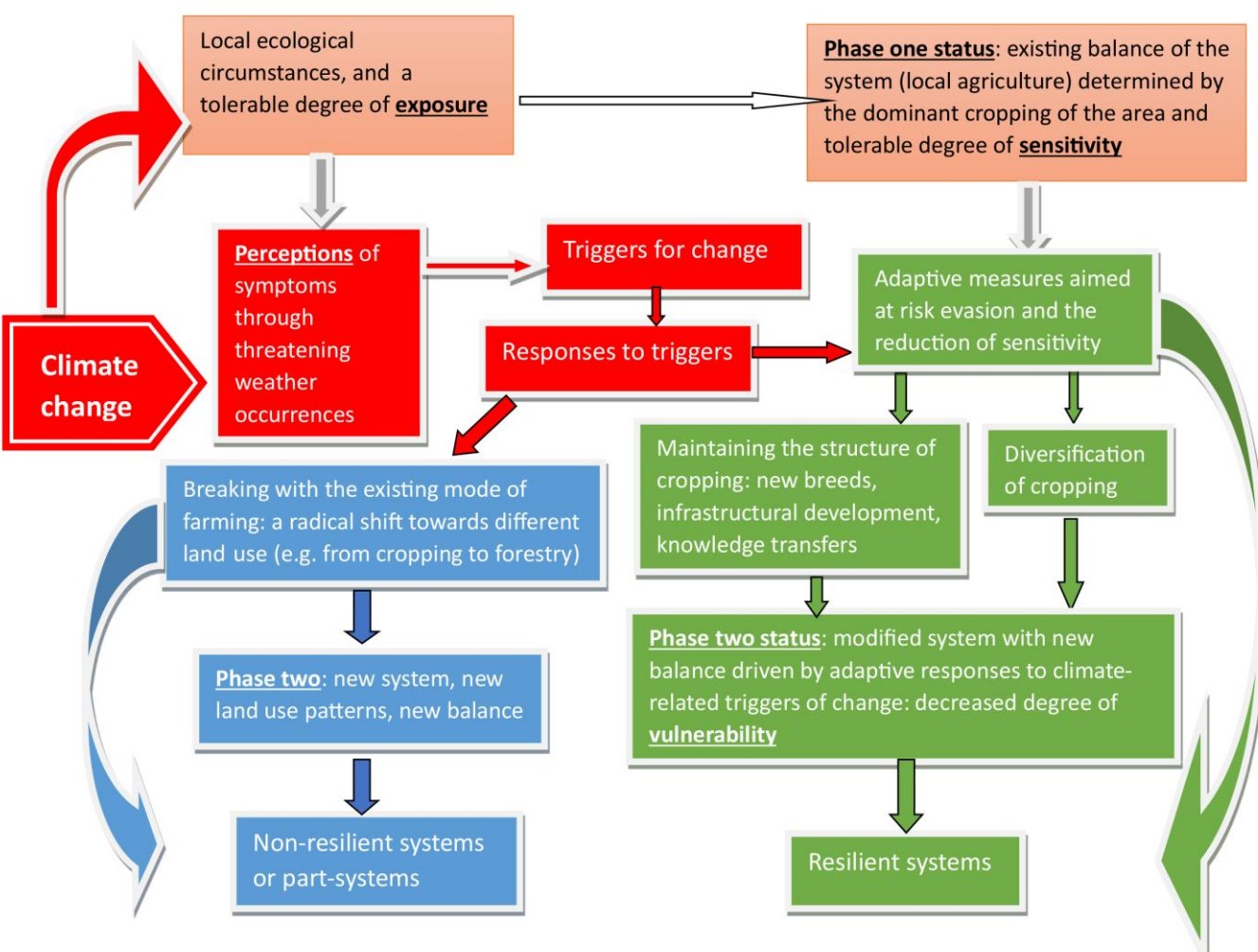

**Figure 5.** Resilient and non-resilient agricultural systems. Source: own elaboration.

Our findings also align with other studies on climate change perception, indicating that farmers often struggle to differentiate between climate variability and climate change, particularly regarding extreme weather events. Some studies suggest that, unlike temperature, farmers' perception of rainfall is also not consistently aligned with meteorological data. In the case of our Hungarian studies, despite varying reductions in precipitation, farmers in each area consistently expressed concerns about insufficient rainfall. However, this apparent contradiction can be resolved by considering that rising temperatures lead to increased evapotranspiration and a heightened susceptibility to drought. Thus, farmers may legitimately feel that the amount of rainfall is no longer sufficient for the established crop and production structure, even if measured precipitation has only slightly decreased.

The interviewees in our case studies exhibited a broad range of adaptive practices across various aspects of agricultural activity. This aligns with previous studies, suggesting a comprehensive exploration of adaptive strategies. Consistent with other research, our results highlight the interconnectedness of climate change exposure with other adaptation pressures, such as those related to market dynamics. However, the responses from the interviewees also reveal that farmers do not perceive these challenges in isolation. Instead, they seek adaptation methods that address multiple challenges simultaneously.

Consistent with previous studies, our paper identified various factors influencing farmers' perception and adaptive capacity. Notably, social networking emerged as the most prominent factor, intertwined with the dissemination of information; this was significantly strengthened by the presence of an active Producer Organization. This observation stands

out as a key takeaway for Hungarian development policy, as a good practice for facilitating climate change adaptation.

As discussed in Section 2, the progression of scientific discussion on this topic has been hindered by loosely formulated and inconsistent definitions of key terms and concepts. One of the key strengths of this paper is its holistic approach, contributing not only to a clearer definition of these related terms but also providing a solid framework for comparing the Hungarian experience to global findings.

Finally, to conclude with some of the inevitable limitations of our investigations and suggestions for future research. The two most prominent limitations in relation to our research were connected to the case studies and data availability. Due to the significant time requirements of field research, we had to limit the number of case studies we could explore. While we endeavored to maximize their diversity, certain areas are unavoidably missing. Notably, we did not include the traditional fruit-growing areas of Szatmár and Bereg on the Northern Hungarian Plain. These regions face substantial reductions in precipitation, exist on the outer periphery with unfavorable socioeconomic dynamics, and have been heavily impacted by the loss of eastern markets and the transition to a market economy. Another area that would be valuable for field examination is Zala county in Western Transdanubia. This region falls within a territory with increased rainfall and a more pronounced Mediterranean influence. Farmers in Zala demonstrate great ingenuity in adaptive practices, such as introducing kiwis or figs into the area.

Data availability emerged as another constraint. The absence of data for certain phenomena, such as social activities, hindered a more nuanced comparison of qualitative and quantitative findings. The sparse network of meteorological stations limited the in-depth examination of small-scale differences in extreme weather events. In some instances, only outdated data, such as information from the Population Census of 2011, were accessible. While the snowballing method proved useful in identifying new potential interviewees, it also resulted in our subjects generally being more deeply embedded in local networks and holding a higher social status than average.

Building on the strengths of the current paper and recognizing its shortcomings identified above, we propose several steps for enhancing future research. Efforts should be directed toward overcoming the constraints of data availability. This could involve tapping into fresh data sources, including the recent census, and exploring new avenues for data collection. The addition of new case studies, for instance, from the Northern Hungarian Plain or Western Transdanubia, would offer additional insights into how local environmental and socioeconomic conditions influence perception and adaptive capacity. Leveraging the solid theoretical framework presented in this paper, a systematic study for East–Central Europe could be developed, allowing for a comparative analysis of adaptation strategies in response to the dual challenges of climate change and market-related pressures.

**Author Contributions:** Conceptualization, J.L., K.K., G.K. and B.K.; methodology, J.L. and K.K.; investigation, J.L., K.K., B.K., C.B., E.H., G.K., K.R., A.D.K. and M.M.V.; resources, J.L.; writing— original draft preparation, J.L., K.K., G.K, B.K. and A.D.K.; writing—review and editing, N.S.; visualization, J.L. and K.K.; project administration, K.K.; funding acquisition, K.K. and B.K. All authors have read and agreed to the published version of the manuscript.

**Funding:** The research project titled 'Farm Types, Challenges, Directions of Adaptation and their impact on the Hungarian Countryside' is funded by the NKFIH (National Research, Development and Innovation Office). Project number: K 132 975. Research for this paper is also supported by the János Bolyai Research Scholarship of the Hungarian Academy of Sciences (BO/00583/22/10—New horizons for agent-based modeling of the spatial processes of Hungary). The APC was funded by NKFIH Project number: K 132 975. We are thankful for the financial support received from NKFIH.

**Informed Consent Statement:** The GDPR consent form was signed by the interviewees. We are grateful to our respondents for their time and trust.

**Data Availability Statement:** The datasets used and/or analyzed during the current study are available from the corresponding author on reasonable request.

**Acknowledgments:** We are grateful to our interview partners for the time they spent with us and for trusting us.

**Conflicts of Interest:** The authors declare no conflict of interest.

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
