# Peer review of "Climate Change, Pressures, and Adaptation Capacities of Farmers: Empirical Evidence from Hungary"

_horticulturae, doi:10.3390/horticulturae10010056_

Round 1

Reviewer 1 Report

Comments and Suggestions for Authors

Thank you for giving me this opportunity to review this paper. This paper is very interesting, and I believe it will be more widely disseminated if it is improved. Individual recommendations are as follows:

1. Some language is debatable. For example, the first sentence (Line 39) and the exclamation point included in line 82. Academic papers rarely appear exclamation marks, please think.

2. The research conclusions of the paper are obvious, but the author lacks discussion about whether these research results verify or overturn the past conclusions compared with predecessors or his own research, and why these research results are so, and some corresponding explanations are needed. This is what the discussion is about.

3. Potential contributions to this paper should be pointed out by the author.

4. The limitations of this article should be pointed out.

5. The author should give the chapter arrangement of the paper in the introduction.

6. The references in this paper are not enough, so it is suggested that the author increase the number to about 45-50, and many useful and authoritative references are not involved.

Author Response

Dear Reviewer,

Thank you for your comments and suggestions. We agree with the proposed changes and have implemented the following improvements:

  • The final section of the paper has been expanded to include a discussion of the results in comparison with the findings of the existing scientific literature;
  • The paper now contains a discussion about the limitations of the article, and a description of chapter arrangement;
  • We significantly increased the number of the references and added an extended literature review;
  • The noted questionable sentences were corrected, and the whole paper was revised by a native speaker.

We also addressed the other mentioned issues. For a detailed answer, please see the attachment. In addition to the responses, the file includes the updated version of the paper, with the revised portions and new additions highlighted in red.

Once again, we sincerely thank you for dedicating your time to assist us in refining our paper.

Best regards,

The Authors

Reviewer 2 Report

Comments and Suggestions for Authors

The research results presented by the authors in Climate Change, Pressures and Adaptation Capacities of Farmers: Empirical Evidence from Hungary”  are interesting suitable for publishing in Horticulturale and probably may be used in further researches and analyses.

My first comment to the article-please specify practical aim of your research and article.

Abstract

 Please add some results of your research in to your abstract.“

Introduction-

line 39-40- please modify language. Rewrite  it.

Hungary is in a special situation in this respect, as the results of regional climate models (PRUDENCE) project an increase in uncertainty: while average temperatures are clearly increasing, precipitation projections differ strongly [4,5].- this is not clear for me.

Explain term PRUDENCE please. The same IPPC

Maybe you can create lists of abbreviations: before chapter 1.

In the end you presented objectives of your study.

Please add also information what was practical and scientifically aim of your study.

Maybe it will be more convenient to create separate chapter 2. called Theoretical framework

Materials and data

I suggest to change the title of this chapter into Materials, methods and data

Chapter 2.1

What was the reason for chosen four districts?

What were the initial assumptions when selecting districts?

Check table 1 superscripted indexes (km2 not  km2 for example.

Figure 3 and 4. Terms Number of heat days and Chance of late spring frosts, Heat days  are not clear

Discussion and results

Separate discussion and results chapters should be created. Briefly state in bullet points the main conclusions of the research carried out.

References

Is it possible to discuss the research better and include more literature references?

I am certain that the authors can improve the form of their article without any problems using my comments.

Comments on the Quality of English Language

Please use native speaker to improve your article.

Author Response

Dear Reviewer,

Thank you for your comments and suggestions. We agree with the proposed changes and have implemented the following improvements:

  • We specified the practical aim of our research;
  • The paper now contains additional information about the selection of the case studies;
  • The figures were modified;
  • The final section of the paper has been expanded to include a discussion of the results in comparison with the findings of the existing scientific literature, as well as the limitations of the research and future research directions;
  • We significantly increased the number of the references and added an extended literature review;
  • The whole paper was revised by a native speaker.

We also addressed the other mentioned issues. For a detailed answer, please see the attached document. In addition to the responses, the file includes the updated version of the paper, with the revised portions and new additions highlighted in red.

Once again, we sincerely thank you for dedicating your time to assist us in refining our paper.

Best regards,

The Authors

Reviewer 3 Report

Comments and Suggestions for Authors

Thank you for the opportunity to cooperate to improve the manuscript entitled Climate Change, Pressures and Adaptation Capacities of Farmers: Empirical Evidence from Hungary, which aims to analyze the climate exposure, sensitivity, perception, adaptive capacity, vulnerability and resilience of the Hungarian agricultural sector, focusing on fruit, vegetable, and grape producers.

Abstract. Objective stated, method informed, results presented. OK

1. Introduction. Past and present studies presented, objective stated. Please include some reference(s) between lines 71-81.

Line 113-123, please include some reference(s)

Line 124-125, please include reference to this definition. 

Line 167-171, you started the citation with one reference [21] and finished informing another one. Please define citation. Direct citations must have the page number provided.

2. Methods and data. Well structured and explained. Good.

3. Results. Fig. 4. Is it possible to amplify to allow the Legend to be read?

Well stated. The results of the interviews were very good.

4. Conclusion and discussion. 

Fig. 5. Is it possible to increase the font size?

Please inform limitations of your study and include suggestions for future studies.

References. Please include some references from Horticulturae in your paper, if possible from 2021 to 2023. 

Author Response

Dear Reviewer,

Thank you for your thoughtful comments and suggestions. We agree with the proposed changes and have implemented the following main improvements:

  • We significantly increased the number of the references and added an extended literature review;
  • The final section now incorporates a discussion on the limitations of the paper and suggests future research directions;
  • The figures have been modified to enhance readability.

We also addressed the other mentioned issues. For a detailed answer, please see the attachment. In addition to the responses, the file includes the updated version of the paper, with the revised portions and new additions highlighted in red.

Once again, we sincerely thank you for dedicating your time to assist us in refining our paper.

Best regards,

The Authors

Round 2

Reviewer 1 Report

Comments and Suggestions for Authors

Thanks to the authors for their efforts. I have no further comment. I propose to publish. Merry Christmas to you all!

Author Response

Dear Reviewer, thank you very much for your contributions in improving the paper. Happy New Year!